# Tool Cache Agent: Accelerating LLM Agent Through Intelligent Tool Call Caching

## Abstract

The rapid advancement of large language models (LLMs) is driving the emergence of LLM agents. Unlike standalone LLMs, these agents interact dynamically with their environment, employing tools, multi-step processes, and even multiple LLMs to enhance functionality. Optimizing tool usage is critical for LLM agents. In this paper, we introduce ToolCacheAgent, an adaptive "agent-for-agents" that automatically caches tool call results to improve response time and reduce redundant computation. For each tool in the agent workflow, ToolCacheAgent generates a caching plan that specifies cacheability, expiration, and inter-tool invalidation rules to maintain correctness in stateful executions. It continuously monitors runtime signals and adapts its cache policies to handle shifting workloads and memory pressure. We evaluate ToolCacheAgent across a range of agent workloads with diverse tool usage patterns and observe up to a $1.69\times$ latency speed-up without compromising accuracy.

## 1 Introduction

Recent advances in large language models (LLMs) have enabled a new class of systems known as LLM agents—systems that actively interact with their environment through tool use, multi-step reasoning, and even collaboration across models. Tools serve as external functions that extend an agent's capabilities, providing access to real-time, domain-specific information beyond the limits of pretraining OpenAI (2023). By integrating perception, planning, and action, LLM agents can reason over complex workflows and carry out goal-directed behavior. Recent studies have demonstrated the growing capabilities of LLMs in performing such tasks Ouyang et al. (2022); Wang et al. (2024); Bubeck et al. (2023); Wei et al. (2022), highlighting their potential for solving real-world problems.

Figure 1(a) illustrates a typical LLM agent workflow. Upon receiving a user query, the agent devises a task execution strategy, calls the necessary tools, and synthesizes a final response. This process distinguishes LLM agents from traditional LLMs in two significant ways: (1) they utilize tools (e.g., external functions) to interact with the real world, and (2) they employ a multi-step execution process that combines tool calls with LLM-generated responses.

Optimizing the performance of LLM agents requires improving the execution graph composed of multiple tool calls and LLM generations. While prior work has primarily focused on enhancing LLM accuracy, throughput, and response latency, the cost of external tool execution has received comparatively less attention. In some workflows, such as those shown in Figure 1(b), tool calls can account for a significant portion of total latency—up to 61% in the Movie Recommendation dataset and over 55% in ParallelQA—when using standard agent architectures like ReAct. These results suggest that, depending on the agent design and workload, tool invocation may become a dominant bottleneck. As agents increasingly incorporate complex toolchains and rely on external APIs or databases, improving tool execution efficiency becomes an important consideration for end-to-end performance.

LLMCompiler Kim et al. (2024b), shown in Figure 1(c), is a recent system that improves agent execution by enabling parallel scheduling of tool calls. However, it re-executes each tool call regardless of whether identical arguments have previously been used, leading to redundant computation. This inefficiency is especially pronounced in scenarios involving expensive tools, such as statistical analytics or large database queries. Without reuse or caching, these redundant executions inflate response time and system load, limiting scalability and responsiveness in real-world deployments.

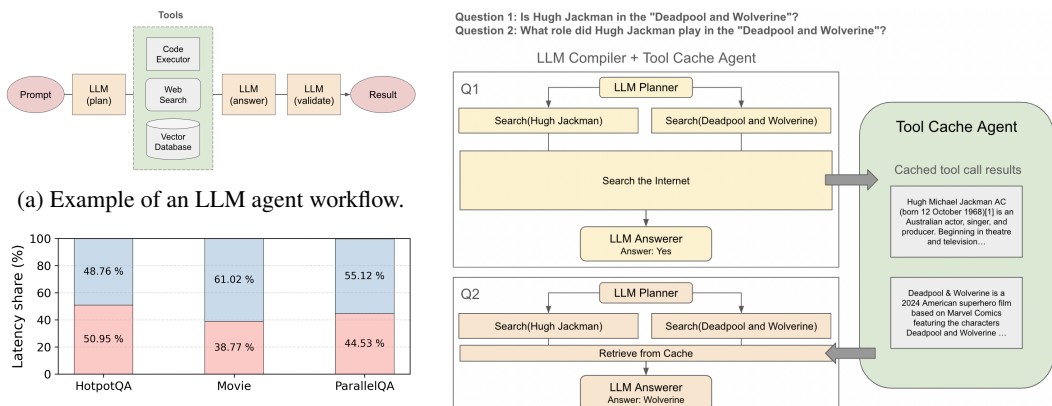

(a) Example of an LLM agent workflow.

(b) Latency breakdown of tasks run on Re-Act

(c) ToolCacheAgent integrated with LLM Compiler.

Figure 1: **(a)** A standard LLM agent architecture that performs task decomposition (planning), invokes external tools (e.g., search, math, code execution), synthesizes answers, and optionally validates the results. **(b)** Latency breakdown of tasks from three datasets—HotpotQA, Movie Recommendation, and ParallelQA—executed using the ReAct agent. The stacked bars show the percentage of total latency attributed to LLM generation, tool calls, and other operations. **(c)** An example of ToolCacheAgent integrated with LLM Compiler. The agent caches tool call results after the first query (Q1), allowing a faster response for the second query (Q2) by retrieving previously computed results from the cache.

To tackle this issue, we propose ToolCacheAgent, an adaptive and autonomous "agent-for-agents" that improves efficiency by intelligently caching and managing tool execution results. Tool-CacheAgent stores the results of idempotent tool calls and retrieves them for subsequent requests with identical arguments, thereby eliminating redundant executions. However, caching tool results introduces unique challenges due to the diverse nature of tools. Some tools, such as mathematical operations, are static and idempotent—ideal candidates for caching. Others, like tools that send emails, must execute every time to ensure correct functionality. To address this, ToolCacheAgent generates caching strategies through an integrated LLM-driven planner and adapts its cache policies in response to workload variations. Specifically, it autonomously determines the cacheability of each tool invocation and calculates optimal parameters such as cache expiration times. Moreover, it introduces novel dependency-aware invalidation rules to ensure cached results remain consistent and accurate when underlying data changes.

To the best of our knowledge, ToolCacheAgent is the first framework to implement caching for tool calls in LLM agents, effectively addressing inefficiencies caused by repeated executions. Our contributions are as follows:

• We introduce ToolCacheAgent, an adaptive caching agent for LLM-based systems that autonomously updates caching strategies in response to runtime performance and workload shifts.

• We design a Cache Planner, a language model–driven module that classifies tools as READ or WRITE, infers their cacheability class (STATIC, TRANSIENT, or NONE), and assigns expiration times for cacheable entries.

• We implement dependency-aware invalidation, a novel mechanism that tracks interactions between tools and invalidates cached outputs when upstream data changes, reducing semantic errors caused by stale reads.

• We demonstrate the end-to-end effectiveness of ToolCacheAgent across three benchmarks—HotpotQA, Movie Recommendation, and ParallelQA—achieving up to $1.69\times$ latency speedup without compromising accuracy. We further validate its adaptive replanning capability on a mixed dataset and its dependency-aware invalidation on the $\tau$-bench Retail benchmark.

## 2 BACKGROUND

### 2.1 LLM AGENTS

Recent advancements in LLM agents have tackled a wide range of challenges across various domains Zhou et al. (2024a); Yang et al. (2024); Hong et al. (2024); Shao et al. (2024). This progress has spurred the development of frameworks such as LangChain, AutoGen, and LlamaIndex, which streamline the creation and deployment of LLM agents Chase (2022); Wu et al. (2023); Liu (2022); Liu et al. (2024). Current research focuses on improving agent performance, particularly in response time and accuracy.

Several studies enhance agents' planning and task management capabilities Huang et al. (2024); Wan et al. (2024); Song et al. (2023), while others refine tool utilization to increase agent functionality Zhang et al. (2024). Self-improvement mechanisms, such as feedback loop-based methods, allow agents to adapt to their environments and optimize performance over time Shinn et al. (2023).

While much research has been conducted on optimizing LLM inference Yu et al. (2022); Leviathan et al. (2023); Cai et al. (2024); Kim et al. (2024a), research into agent-level optimization remains limited. LLMCompiler parallelizes function calls automatically to enhance efficiency. Recent studies model agents using traditional computer systems to achieve optimization Karpathy (2023); Packer et al. (2024); Singh et al. (2024); Zhou et al. (2024b). Complementing these efforts, this study investigates agent-level caching to minimize redundant computations and enhance agent performance.

### 2.2 LLM CACHING

Caching LLM-generated outputs reduces redundant computations and improves overall efficiency. In LangChain, keyword caching checks whether a cached output exists for an identical input Chase (2022). If a match is found, the system immediately returns the cached result; otherwise, the input is processed. While straightforward, this approach is limited to exact matches, reducing its broader applicability.

Semantic caching addresses this limitation by caching results based on query meanings, often using vector embeddings Bang (2023). This extends caching to semantically similar queries, broadening its applicability. However, it risks returning inaccurate results when semantic similarity does not ensure equivalence.

Another approach focuses on caching attention states during input processing Gim et al. (2024); Hu et al. (2024). By reusing these cached states, it reduces prefill-stage computation and accelerates text generation for queries with overlapping prefixes. Similarly, in retrieval-augmented generation (RAG), caching document-level results for repeated queries enhances efficiency Jin et al. (2024).

While existing research primarily addresses caching mechanisms for LLM text generation, this work extends caching to tool calls within agent workflows. By minimizing redundant tool executions, this approach improves agent efficiency and performance.

## 3 TOOLCACHEAGENT: DESIGN AND COMPONENTS

ToolCacheAgent is an LLM agent designed to reduce redundant tool executions in existing agent workflows through selective caching. It supports modular integration, requiring minimal modifications to existing systems while remaining adaptable to dynamic runtime behavior. Figure 2 provides an overview of ToolCacheAgent's main components, which include the Cache Manager, Historical Database, Cache Planner, and Orchestrator.

### 3.1 CACHE MANAGER

The Cache Manager handles the storage, retrieval, and eviction of cached tool execution results. It constructs unique cache keys based on tool names and argument hashes, enabling fast and precise lookups. Configurable parameters such as `max_memory` and `eviction_policy` define cache size constraints and eviction behaviors (e.g., Least Recently Used, LRU). Cache entries are stored and evicted according to these parameters.

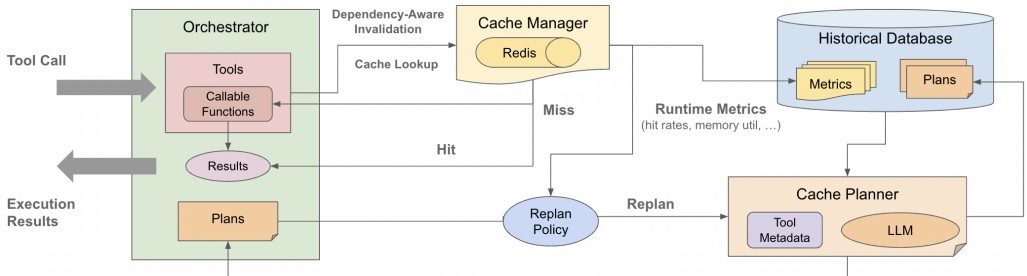

Figure 2: System architecture of ToolCacheAgent. The orchestrator intercepts tool calls and consults the Cache Manager to determine whether a cached result exists. On a cache hit, the result is returned immediately; on a miss, the tool is executed and its result is stored. The Cache Planner uses tool metadata, runtime metrics, and previous plans stored in the Historical Database to generate caching plans via an LLM. A Replan Policy module monitors hit rates, eviction events, and memory pressure, triggering replanning when performance degrades.

## 3.2 HISTORICAL DATABASE

The Historical Database component persistently records cache usage metrics, historical cache plans, and decisions made by the Cache Planner. It logs runtime statistics such as cache hit rates, eviction rates, and memory utilization. This historical information supports informed decision-making by the Cache Planner, enabling accurate and context-aware planning and replanning.

## 3.3 CACHE PLANNER

The Cache Planner is responsible for determining the cacheability of tool calls and generating appropriate cache policies. It begins by assessing each tool's characteristics, such as determinism and output stability, to establish suitable caching parameters. It leverages the power of an integrated LLM planner—guided by structured prompts and historical metrics stored in the Historical Database—to generate detailed cache plans. Each plan explicitly defines cacheability status, recommended time-to-live (TTL), and potential eviction behaviors for dependent tools.

## 3.4 ORCHESTRATOR

The Orchestrator acts as the central controller within ToolCacheAgent, coordinating interactions among the Cache Manager, Historical Database, and Cache Planner. Upon receiving tool calls, the Orchestrator first queries the Cache Manager to determine if a result is already cached. If so, it immediately returns the stored result; otherwise, it executes the tool and caches the result according to the current cache plan. The Orchestrator also periodically triggers adaptive replanning based on defined conditions.

## 4 PLANNING AND ADAPTIVE REPLANNING STRATEGY

Caching tool outputs in LLM agent workflows presents unique challenges, as tools vary widely in their behavior and side effects. While some functions are deterministic and stateless, others produce non-repeatable effects or depend on external context, making them unsuitable for naive caching. To address this, ToolCacheAgent employs an integrated LLM-based planner that generates tailored caching strategies based on tool semantics and observed runtime behavior. The planner continuously adapts cache parameters—such as expiration times and eligibility—to maintain correctness while improving efficiency under changing workloads.

## 4.1 CACHE PLANNING MECHANISM

The Cache Planner determines caching strategies by analyzing tool metadata and runtime behavior. During the initial planning phase, only static metadata—such as tool names and descriptions—is

available. The planner uses this information to generate preliminary cache plans. In contrast, subsequent replanning phases incorporate execution statistics and previous plans from the Historical Database to refine cacheability decisions and tune parameters such as expiration times.

Each cache plan comprises a set of per-tool directives, where each tool is explicitly classified as either READ or WRITE. This distinction determines which fields are applicable. For READ tools, the planner assigns a cacheability label (STATIC, TRANSIENT, or NONE), an optional expiration time, and a list of primary arguments that influence cache key construction.

For WRITE tools, the plan may include a list of invalidation rules that define how the tool affects cached results of related READ tools. These fields enable ToolCacheAgent to build precise keys and coordinate safe invalidation. The schema-level structure is shared across plans, but the semantics of these fields—particularly primary arguments and invalidation rules—are described in detail in Subsection Dependency-Aware Invalidation.

Plan generation is driven by structured LLM prompts that include detailed instructions and a strict JSON schema. The planner produces heuristically informed caching policies tailored to each tool's semantics. For instance, deterministic functions such as mathematical computations may be classified as READ + STATIC, whereas tools that reflect dynamic or external state—such as real-time APIs—may be marked as TRANSIENT with short TTLs or excluded from caching entirely.

The planner applies a two-step structured reasoning process:

1. It first generates intermediate "thoughts" based on tool descriptions and, when available, runtime metrics, outlining the rationale behind each caching decision and any inferred dependencies.

2. It then converts these into executable cache plans, expressed in structured JSON, which encode all necessary information to generate keys, invalidate dependencies, and enforce TTLs.

## 4.2 DEPENDENCY-AWARE INVALIDATION

Many tools exhibit implicit dependencies: the output of one tool becomes stale when another modifies overlapping state. To maintain correctness under such conditions, ToolCacheAgent introduces *dependency-aware invalidation*. This mechanism allows the planner to explicitly declare cache invalidation rules that capture inter-tool dependencies at the parameter level.

Each cacheable READ tool declares a set of *primary arguments*—parameters that affect output freshness and appear directly in cache keys. Conversely, any WRITE tool may define *invalidation rules*, each mapping its own arguments to the primary arguments of a related READ tool. When the WRITE tool executes, ToolCacheAgent constructs a deterministic cache-key prefix from the matching arguments and issues a prefix-based invalidation to evict affected entries.

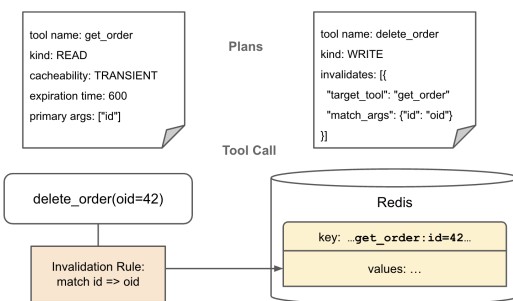

Figure 3: Example of dependency-aware invalidation. The planner marks get_order as a READ tool with TRANSIENT cacheability and "id" as its primary argument. The delete_order tool is marked as WRITE and includes an invalidation rule targeting get_order. When executed, the agent derives a key prefix from the matching "id" value and evicts the affected cache entries.

Figure 3 illustrates an example of dependency-aware invalidation. Consider a tool get_order(id) whose results are cached using id as a primary argument. A corresponding tool, delete_order(oid), would include an invalidation rule stating that oid maps onto the primary arguments of get_order. Upon execution, any cache key with the matching prefix get_order:id=<val> is removed, ensuring stale results are not served.

This structured invalidation logic is generated automatically by the planner and embedded within each cache plan. It enables ToolCacheAgent to maintain strong cache correctness guarantees across multi-tool workflows without sacrificing efficiency.

### 4.3 ADAPTIVE REPLANNING

Caching decisions that perform well under one workload may degrade performance as the input distribution shifts or memory fills up. ToolCacheAgent addresses this by augmenting the initial *planning phase* with a lightweight, online *replanning mechanism*. Replanning is triggered only when runtime telemetry indicates a likely performance regression.

**Runtime Signals.** At periodic intervals (e.g., every $N = 100$ tool calls), the orchestrator samples a set of cache-level metrics:

**Hit ratio ($h_t$):** Fraction of cache hits within the current window, where $h_t \in [0, 1]$.

**Evictions ($e_t$):** Number of keys evicted during the same interval.

**Memory usage ($u_t$):** Utilization, defined as $u_t = \mathtt{used\_memory}_t / \mathtt{maxmemory}$, with $u_t \in [0, 1]$.

To reduce short-term volatility, we compute exponentially weighted moving averages (EWMA):

$$\hat{h}_t = \alpha h_t + (1 - \alpha)\hat{h}_{t-1}, \quad \hat{e}_t = \alpha e_t + (1 - \alpha)\hat{e}_{t-1}, \tag{1}$$

where $\alpha \in (0, 1]$ is a smoothing constant. After each plan is installed, the baseline values $H_0 = \hat{h}_{t_{\text{last}}}$ and $E_0 = \hat{e}_{t_{\text{last}}}$ are recorded for comparison.

**Trigger Condition.** Replanning is attempted at time $t$ if the following predicate holds:

$$\hat{h}_t < (1 - \theta_H)H_0 \quad \text{or} \quad \hat{e}_t > E_0 + \theta_E(1 - \beta u_t), \tag{2}$$

where $\theta_H$ and $\theta_E$ are threshold constants, and $\beta$ controls sensitivity to memory pressure.

The first clause triggers when the smoothed hit ratio drops significantly below the baseline. The second clause captures eviction spikes under memory pressure. As $u_t$ approaches 1 (i.e., cache nearly full), even modest increases in evictions can justify replanning. This coupling ensures the system only replans when memory is saturated *and* eviction activity increases. Unless otherwise stated, we use: $\alpha = 0.5$, $\theta_H = 0.10$, $\theta_E = 50$, and $\beta = 2.0$. The policy requires only four scalar variables and performs $O(1)$ arithmetic per sample, which is negligible compared to cache lookups or tool executions.

## 5 EVALUATION

We evaluate the effectiveness of ToolCacheAgent across a range of agents, models, and tool-using benchmarks. The evaluation is structured into three parts, each corresponding to a core capability of our system. First, in *End-to-End Gains from Caching*, we assess how ToolCacheAgent improves latency and throughput across different agents and memory budgets, using both fixed and constrained cache settings. Next, in *Adaptive Replanning under Workload Shift*, we demonstrate how ToolCacheAgent dynamically adjusts its cache policy in response to shifts in workload composition, reducing performance degradation through a versioned namespace mechanism. Finally, in *Dependency-Aware Invalidation*, we evaluate ToolCacheAgent's ability to maintain correctness under stateful tool usage by explicitly invalidating cached reads after writes. Each subsection includes the relevant experimental setup and results.

### 5.1 END-TO-END GAINS FROM CACHING

We begin by evaluating the end-to-end performance benefits of ToolCacheAgent when integrated into two tool-using agents: LLMCompiler and ReAct. Experiments were conducted on two NVIDIA H100 GPUs, with a local Redis instance serving as the cache backend. For each configuration, we ran three independent trials and report the average results. Two open-weight language models—Llama 3.3 70B and Qwen 2.5 72B—were tested across three datasets reflecting diverse tool usage characteristics.he replanning trigger condition was evaluated every 120 requests to decide whether cache plans should be updated during execution.

Table 1: End-to-end accuracy, latency, and speed-up across three benchmarks with and without Tool-CacheAgent (TCA). We evaluate two agents (ReAct and LLMCompiler) under two memory budgets (100% and 50%) on two model backends: Llama-3.3 70B and Qwen-2.5 72B. ToolCacheAgent consistently reduces latency without harming accuracy.

| Dataset | Method | Llama-3.3 70B | | | Qwen-2.5 72B | | |
|---|---|---|---|---|---|---|---|
| | | Accuracy | Latency (s) | Speed-up | Accuracy | Latency (s) | Speed-up |
| HotpotQA | ReAct | 0.70 | 6.13 | – | 0.68 | 6.17 | – |
| | ReAct+TCA-100% | 0.70 | 5.93 | 1.03× | 0.68 | 5.83 | 1.06× |
| | LLMCompiler | 0.68 | 3.78 | – | 0.65 | 3.63 | – |
| | LLMCompiler+TCA-100% | 0.68 | 3.64 | 1.04× | 0.65 | 3.58 | 1.01× |
| Movie Rec. | ReAct | 0.77 | 21.31 | – | 0.76 | 14.43 | – |
| | ReAct+TCA-100% | 0.77 | 13.15 | 1.62× | 0.76 | 8.71 | 1.66× |
| | ReAct+TCA-50% | 0.77 | 13.96 | 1.53× | 0.76 | 9.02 | 1.60× |
| | LLMCompiler | 0.82 | 6.52 | – | 0.70 | 7.28 | – |
| | LLMCompiler+TCA-100% | 0.82 | 5.51 | 1.18× | 0.70 | 6.19 | 1.18× |
| | LLMCompiler+TCA-50% | 0.82 | 5.64 | 1.16× | 0.70 | 6.34 | 1.15× |
| ParallelQA | ReAct | 0.84 | 25.35 | – | 0.90 | 19.29 | – |
| | ReAct+TCA-100% | 0.84 | 15.01 | 1.69× | 0.90 | 14.47 | 1.33× |
| | ReAct+TCA-50% | 0.84 | 16.79 | 1.51× | 0.90 | 16.05 | 1.20× |
| | LLMCompiler | 0.82 | 7.43 | – | 0.88 | 7.98 | – |
| | LLMCompiler+TCA-100% | 0.82 | 6.17 | 1.20× | 0.88 | 6.54 | 1.20× |
| | LLMCompiler+TCA-50% | 0.82 | 6.48 | 1.15× | 0.88 | 7.03 | 1.14× |

Table 2: Cache hit rates and eviction counts for ReAct and LLMCompiler across three datasets under full (100%) and constrained (50%) memory budgets. Eviction counts are omitted when the full memory budget could hold all entries.

| | | Llama-3.3 70B | | | Qwen-2.5 72B | | |
|---|---|---|---|---|---|---|---|
| | | 100% mem | 50% mem | | 100% mem | 50% mem | |
| Dataset | Agent | Hit% | Hit% | Evict | Hit% | Hit% | Evict |
| HQA | ReAct | 6.8 | - | - | 10.1 | - | - |
| | LLMC | 9.4 | - | - | 7.9 | - | - |
| Movie | ReAct | 60.8 | 57.2 | 639 | 74.4 | 70.3 | 253 |
| | LLMC | 61.7 | 58 | 412 | 61.6 | 57.6 | 377 |
| PQA | ReAct | 43.1 | 31.5 | 539 | 43.9 | 26.8 | 591 |
| | LLMC | 40.9 | 32 | 509 | 42.8 | 31.7 | 517 |

Table 3: Effect of dependency-aware invalidation on correctness and cache efficiency.

| Metric | w/o Invalidation | w/ Invalidation |
|---|---|---|
| Wrong results (out of 582) | 35 | 6 |

| Tool | Hits | Misses | Hit ratio | TTL(s) |
|---|---|---|---|---|
| find_user_id_by_name_zip | 35 | 27 | 0.565 | STATIC |
| get_order_details | 53 | 118 | 0.310 | 3600 |
| get_product_details | 44 | 29 | 0.603 | 3600 |
| list_all_product_types | 5 | 1 | 0.833 | STATIC |
| get_user_details | 30 | 29 | 0.508 | 3600 |
| find_user_id_by_email | 8 | 7 | 0.533 | STATIC |
| calculate | 0 | 14 | 0.000 | STATIC |

**Memory Budget Settings.** We vary the cache memory budget across two regimes: (i) 100% budget, where the cache is large enough to store all intermediate tool outputs without evictions, and (ii) 50% budget, where the cache is limited to half the total tool call footprint, leading to evictions. These configurations simulate best-case and constrained scenarios, respectively. All datasets were evaluated in the original order provided, without shuffling or reordering.

**Datasets.** **HotpotQA** Yang et al. (2018) contains 1.5K comparison-style multi-hop questions. The 2,671 unique search queries exhibit low locality (average reuse: 1.11, std-dev: 0.34, min: 1, max: 4). **Movie Recommendation**, from the Beyond-the-Imitation-Game Benchmark Aarohi Srivastava (2023), includes 500 examples that request the most similar movie from a candidate set. This dataset is highly skewed in its tool usage: among 148 unique search queries, a small number dominate the request distribution (average reuse: 13.55, std-dev: 24.04, min: 1, max: 125). This skewed pattern enables high hit rates even under constrained memory. **ParallelQA** Kim et al. (2024b) consists of 113 questions requiring sequential tool use, where calculate depends on the output of search. It presents moderate locality with 58 unique search queries (average reuse: 7.19, std-dev: 3.12, min: 1, max: 12), showing more uniform usage than Movie Recommendation. For the calculate tool, it presents low locality with 471 unique queries (average reuse: 1.11, std-dev: 0.37, min: 1, max: 4). These statistics are computed from the datasets themselves, so the distribution and locality of tool queries may vary across different LLMs.

**Main Results.**   Table 1 reports accuracy, latency, and speed-up for each agent, model, and memory budget, with and without ToolCacheAgent. ToolCacheAgent consistently reduces latency while preserving accuracy across all benchmarks. Under full-memory settings, the best-case speed-up reaches $1.69\times$ on ReAct with Qwen 2.5 for ParallelQA, and $1.66\times$ on Movie Recommendation. Even under constrained memory (50%), we observe strong gains—e.g., $1.60\times$ on ReAct with Qwen for Movie Recommendation—demonstrating resilience under pressure. We omit 50% results for HotpotQA due to its extremely low cache hit rate (typically below 10% across runs), which rendered additional experiments uninformative.

**Cache Behavior.**   To understand performance gains in context, Table 2 summarizes cache hit rates and eviction counts. As expected, tasks with frequent tool reuse achieve higher hit rates, enabling greater speed-up. In Movie Recommendation, despite aggressive memory constraints (50%), hit rates remain high due to the skewed distribution of tool calls: dominant keys are repeatedly accessed and remain cached. In contrast, ParallelQA—while benefiting from reuse—shows more sensitivity to memory pressure, reflecting its less skewed distribution of tool inputs. HotpotQA remains below a 10% hit rate across all runs, offering limited caching opportunities. Evictions occur only under the 50% budget and are managed by ToolCacheAgent's TTL and grouping strategies.

These results confirm that ToolCacheAgent improves tool-augmented agent performance end-to-end, particularly when tasks exhibit repeated tool usage.

## 5.2   ADAPTIVE REPLANNING UNDER WORKLOAD SHIFT

To evaluate ToolCacheAgent's ability to adapt under shifting cache demands, we construct a mixed workload from three datasets—HotpotQA, Movie Recommendation, and ParallelQA—using a four-block structure: 100 Movie, 200 HotpotQA, 113 ParallelQA, and 100 Movie-2 requests. All requests are executed using the ReAct agent framework. To accurately reflect the differing reuse patterns and characteristics of each dataset, we treat their tools as distinct: Movie_Search, HQA_Search, PQA_Search, and Calculate are defined as separate tools with independent caching behavior. We test under a constrained 25% memory budget and compare static caching against ToolCacheAgent with online replanning triggered every 120 requests. All cache plans and replanning decisions are generated using the Qwen 2.5 72B language model as the planner.

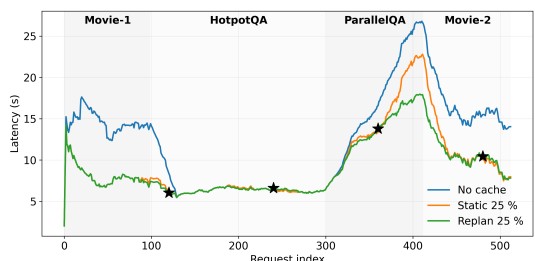

Figure 4: Rolling mean latency under a workload shift (window size = 30). The mixed evaluation stream has four blocks: 100 Movie, 200 HotpotQA, 113 ParallelQA, and 100 Movie-2. All requests use the ReAct agent. We compare no cache (blue), static 25% (orange), and ToolCacheAgent 25% with online replanning (green). Stars mark each replanning event.

As shown in Figure 4, both the static and adaptive variants perform equally well during the initial Movie block, where high hit rates are easily achieved. In the subsequent HotpotQA block, however, all policies converge in latency since Hotpot queries exhibit little to no reuse. Although the first workload shift is detected, replanning is not triggered—overall hit rates remain high due to sustained reuse from the Movie_Search tool. At the second and third events, ToolCacheAgent adapts dynamically: it drops the underutilized Calculate tool and shortens TTLs to counter rising memory pressure and falling reuse. In this experiment, expiration times are discretized into TTL buckets of $\{3600, 600, 60\}$, and the planner demotes HQA_Search across all three levels as its cacheability deteriorates. This controlled TTL decay helps reduce churn and stabilizes cache occupancy. Removing stale Calculate entries also reclaims space for more frequently accessed tools, particularly PQA_Search. The resulting selectivity improves hit rates and lowers latency throughout the ParallelQA phase. At the fourth event, a replan is triggered due to high memory pressure, but no changes to the existing plan were made.

Overall, ToolCacheAgent achieves a mean latency of 8.99s, compared to 9.42s (static) and 12.44s (no cache). The adaptive TTL and tool-level eviction control yield consistent performance gains with no accuracy loss (0.73 across all variants).

## 5.3 DEPENDENCY-AWARE TOOL CACHING

We evaluated ToolCacheAgent on the retail subset of the $\tau$-bench Yao et al. (2024) benchmark, using the GPT-4o trajectory trace. From the 115 tasks in this trace, we extracted the sequence of tool interactions from the `actions` field, yielding a total of 582 tool calls across 15 unique tools. The cache plans were generated using the Qwen 2.5 72B language model as the planner.

**Correctness and Cache Efficiency** ToolCacheAgent successfully identified the role (READ or WRITE) and cacheability class (STATIC, TRANSIENT, or NONE) for all 15 tools in the trace, and generated valid cache plans that guided runtime caching and invalidation decisions. To evaluate the impact of dependency-aware invalidation on correctness, we executed the full trace twice: once without invalidation and once with it enabled. Without invalidation, 35 tool calls returned incorrect results; enabling invalidation reduced this to 6—a $5.8\times$ reduction in error rate. Table 3 presents the per-tool hit ratios and global cache metrics of the 7 READ tools. These results highlight the importance of invalidating stale entries to maintain correctness, particularly for tools that depend on frequently updated state.

**Latency Caveat** Because $\tau$-bench executes tool calls against in-memory mock objects loaded from local JSON files, baseline tool latency is unrealistically low. In our run, the average per-call latency *without* caching was 6.7 ms, while *with* caching it increased to 714 ms—primarily due to Redis lookup overhead dominating the otherwise trivial tool runtime. We emphasize that in real-world deployments where tools involve database or network access, caching is likely to yield significant latency reductions. Our setup was useful for correctness validation but underestimates the latency benefits of caching in practical scenarios.

**Limitations** Despite the improvement, 6 incorrect results remained due to *hidden dependencies*—cases where a WRITE tool modifies state not directly represented in its argument list. For instance, the tool `modify_pending_order_items` internally updates user state linked to the order but does not take `user_id` as an argument. Consequently, cache entries for tools like `get_user_details(user_id)` may remain stale. This exposes a key limitation of argument-based invalidation: it cannot capture latent or indirect dependencies between tools. Several enhancements could address this gap. One option is to augment the planner input with tool implementation code, enabling the planner to infer side effects beyond declared arguments. Alternatively, developers could explicitly annotate tools with side-effect metadata, specifying which entities are affected. Another promising direction is return-value-based invalidation, where the result of a WRITE tool is used to dynamically determine which cached READ entries to purge. These extensions would increase the coverage of invalidation and further reduce the risk of stale reads in complex workflows.

## 6 CONCLUSION

We presented ToolCacheAgent, an adaptive caching framework that improves the efficiency and correctness of LLM agents by automatically caching tool call results. It classifies tools as READ or WRITE, infers cacheability, and generates caching policies—including expiration and invalidation—via an LLM-driven planner. To ensure consistency in stateful workflows, it introduces dependency-aware invalidation and adapts its strategy based on runtime signals such as workload shifts and memory pressure.

Our experiments demonstrate that ToolCacheAgent consistently improves performance across a variety of tool-augmented agent workflows. It achieves up to a $1.69\times$ reduction in latency without compromising accuracy on standard benchmarks. These results establish tool call caching as a critical yet underexplored optimization layer in LLM agents and position ToolCacheAgent as a foundation for future work on efficient, adaptive agent systems.

ETHICS STATEMENT

We used Large Language Models to polish the writing—grammar, wording, and clarity—after the technical content, methods, and results were authored by us. Outputs were reviewed and edited by the authors.

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

# APPENDIX

This appendix provides additional details supporting the main paper, including extended descriptions of our method, implementation details, experimental configurations, and supplementary results.

## A  CACHE PLANNER PROMPT SUITE

### OVERVIEW

The **CachePlanner** uses a two-stage prompting pipeline plus a degradation–aware "re-plan" prompt:

**Step 1:** *Tool Classification ("Thoughts")* – the LLM inspects each tool and decides whether it is a READ/WRITE, lists freshness-determining parameters, and assigns STATIC / TRANSIENT / NONE cacheability classes.

**Step 2:** *Plan Generation* – combines the "Thoughts" from Step 1 with live Redis metrics, the previous plan, and a strict JSON schema to emit a self-consistent cache plan.

**Step 3:** *Re-plan* – invoked only when hit-rate degrades or eviction pressure rises; it analyses metric trends and adjusts TTLs or invalidation edges accordingly.

### A.1  STEP 1 – TOOL CLASSIFICATION PROMPT ("THOUGHTS")

The prompt instructs the model to:

- distinguish READ vs. WRITE tools,
- for each READ, enumerate `primary_args` that govern freshness and choose a cacheability class,
- for each WRITE, list the READs it invalidates and map argument names.

```
You are an expert caching system engineer.
```

```
Examine each tool definition and write a single "Thoughts" section:
  1. Whether it *reads* state or *writes* state.
  2. If READ: list primary_args and choose STATIC / TRANSIENT / NONE.
  3. If WRITE: list which READ tools are invalidated and how.
Return only the Thoughts block, no JSON.

Tool definitions:
{tool_defs}
```

## A.2 STEP 2 – PLAN GENERATION PROMPT

Inputs are (i) tool specs, (ii) live metrics snapshot, (iii) the previous plan, (iv) the JSON schema, and (v) the Step-1 Thoughts. Key guidelines are:

a) start from the previous plan but override entries with low hit ratio *and* high eviction pressure,
b) set primary_args for every READ tool,
c) choose STATIC/TRANSIENT/NONE and TTL (if transient),
d) leave invalidates empty for READs, but fill it for every WRITE.

The model must output *exactly one* JSON object conforming to the schema; prose is rejected.

```
You are **CachePlanner**, tasked with producing a revised cache plan
(JSON only) that follows the schema below.
--Runtime Evidence--
Current Redis metrics snapshot:
{metrics_block}
Previous cache plan (for reference):
{prev_plan_block}
--Tool Specs--
{tools_block}
--Your Thoughts--
{thoughts}
JSON Schema (must match):
{schema_block}

Guidelines
----------
* Start from the previous plan but feel free to override entries whose
  hit ratio is low *and* eviction pressure is high referring to your "
      Thoughts".
* For every READ tool:
  - Set `primary_args` (1) in the correct order.
  - Choose cacheability class. If TRANSIENT, choose a TTL (`
      expiration_time`).
  - The `invalidates` should be empty for a READ tool
* For every WRITE tool:
  - Fill `invalidates` with one rule per affected READ tool.
    Each rule lists which WRITE-call args map onto that READ tool's
    `primary_args`. Omit fields not allowed by the schema.
* Leave `expiration_time` null for STATIC and NONE.
```

## A.3 STEP 3 – DEGRADATION RE-PLAN PROMPT

Triggered when the adaptive policy signals performance drift, this prompt provides:

- the complete history of per-interval metrics,
- the current rolling-window snapshot,
- the existing cache plan.

The model must (i) diagnose trends, (ii) decide per-tool actions KEEP / LENGTHEN_TTL / SHORTEN_TTL / DROP, and (iii) add missing invalidation edges.

```
You are a senior caching-system engineer called in to **re-plan** the
    cache
because performance has degraded.

## Inputs
**Static tool definitions**
(including description and parameters):
{tool_def_block}

**Previous cache plan (excerpt)**
(TTL bucket per tool, invalidation map):
{prev_plan_block}

**Previous metrics history**
Timestamped snapshots taken at each periodic poll since the last plan
(e.g., growing `evicted_keys`, falling hit-rate):
{prev_metrics_block}

**Live metrics (current rolling window)**
{metrics_block}

**Per tool hit ratio**
Timestamped per tool hit ratio snapshots taken at each periodic poll
    since the last plan
{per_tool_metrics}

### Memory-metric glossary
- `u_t` = `used_memory` / `maxmemory` (0\1).
- `evicted_keys` = cumulative evictions since last plan.
- `keyspace_hits` / `keyspace_misses`  raw hit ratio.
- `total_eviction_exceeded_time` = seconds spent above `maxmemory`.

## Task
1. Compare the *trend* in **Previous metrics history** with the **Live
    metrics**.
   Is hit-rate deteriorating steadily?
   Is `u_t` climbing toward 1.0 or stabilising?
   Is the eviction rate accelerating? Briefly diagnose the cause.

2. For every READ tool, pick **KEEP / LENGTHEN_TTL / SHORTEN_TTL /
    DROP**.
  - If memory pressure is high (`u_t  0.9` or evictions rising),
     SHORTEN_TTL, or DROP.
  - If memory pressure is low but hit-rate is falling, consider
     LENGTHEN_TTL for warm tools that may be prematurely evicted.

3. For every WRITE tool, inspect stale-read counts (from hit/miss +
    stale in the metrics).
   Add missing invalidation edges when stale>0.

4. Output a single **Thoughts** section:
   Bullet per tool: *action  one-line justification* (cite hit \%,
     evict \%, or memory trend).
   Extra bullets for any new invalidation edges.
   Final bullet: one-sentence summary of memory status (e.g., u_t =
     0.93 and rising; demoted 5 cold keys to 60 s).
```

## A.4 CACHE-PLAN JSON SPECIFICATION

The planner's output is a single JSON object—`cache_plans`—whose `entries` field forms an ordered list of directives. Each entry describes either a READ or WRITE tool:

- **READ tools** include `cacheability` (NONE, STATIC, TRANSIENT), `primary_args` (names that must appear verbatim in the key), and—when transient—an integer `expiration_time`.
- **WRITE tools** omit those fields but provide `invalidates`—a list of rules that purge stale entries from one or more reader caches. Each rule specifies the `target_tool` (the reader to purge) and an `arg_map` that aligns the writer's arguments with the reader's `primary_args`.

The outer object also records a UTC timestamp (`created_at` RFC-3339) so downstream components can detect and reject stale plans.

```
[Cache-Plan JSON Schema]
{
  "type": "object",
  "name": "cache_plans",
  "description": "Dependency-aware cache plan generated by CachePlanner
     .",
  "properties": {
   "created_at": {"type": "string", "format": "date-time"},
    "entries": {
      "type": "array",
      "items": {
        "type": "object",
        "required": ["tool_name", "kind"],
        "properties": {
          "tool_name": {"type": "string"},
          "kind": {"type": "string", "enum": ["READ", "WRITE"]},

          //  READ-specific
          "cacheability": {
            "type": ["string", "null"],
            "enum": ["NONE", "STATIC", "TRANSIENT", null]
          },
          "primary_args": {"type": "array", "items": {"type": "string"}},
          "expiration_time": {"type": ["integer", "null"], "minimum": 0},

          //  WRITE-specific
          "invalidates": {
            "type": "array",
            "items": {
              "type": "object",
              "required": ["target_tool", "arg_map"],
              "properties": {
                "target_tool": {"type": "string"},
                "arg_map": {
                  "type": "object",
                  "minProperties": 1,
                  "additionalProperties": {"type": "string"}
                }
              }
            }
          }
        },
        "additionalProperties": false
      }
    }
  },
  "required": ["created_at", "entries"],
```

```
  "additionalProperties": false
}
```

This schema is enforced at runtime with `pydantic`; any deviation causes the planner's output to be rejected, ensuring downstream cache components receive a type-safe, dependency-aware plan.

## B    MEMORY CONSTRAINTS EXPERIMENTS

For experiments involving memory constraints, we used Redis's `used_memory` metric to measure memory consumption. The memory usage was recorded twice:

- Before inserting any data into the Redis database.
- After all data for the given dataset had been inserted.

The total memory consumed by the data was calculated by subtracting the initial memory usage from the final memory usage:

$$\text{Memory Consumed} = \texttt{used\_memory\_final} - \texttt{used\_memory\_initial}$$

This method provided a more accurate measure of memory consumption compared to estimating based on raw data size.

## C    ADAPTIVE CACHE REPLAN POLICY

Our cache manager periodically decides whether to rebuild its plan by calling a `shouldReplan()` routine. The decision combines smoothed statistics, instantaneous guards, and a cool-down window to avoid thrashing.

### C.1    PARAMETERISATION

The policy is governed by five tunable parameters:

$\theta_H$    Maximum tolerated *relative* drop in EWMA hit-ratio (default 0.10).

$\theta_E$    Slack on the EWMA eviction delta before it is considered a spike (default 50).

$\beta$    Extra weight on memory pressure when evaluating eviction spikes (default 2.0).

$\alpha$    EWMA smoothing factor (default 0.5).

**cooldown**    Minimum time between consecutive replans (optional).

### C.2    DECISION PROCEDURE

At each monitoring interval the policy receives the current timestamp $t$ and a snapshot of cache metrics. The logic unfolds as follows:

1. **Instantaneous guard.** If the raw hit-ratio falls below $20\%$, replan immediately.
2. **Eviction delta.** Compute the number of new evicted keys since the previous tick.
3. **Memory pressure.** Let $u = \frac{\texttt{used\_memory}}{\texttt{maxmemory}}$ to modulate eviction sensitivity.
4. **EWMA update & predicates.** Update smoothed hit-rate $\hat{h}$ and eviction delta $\hat{e}$ using $\alpha$. A replan is requested when

$$\hat{h} < (1 - \theta_H)\, h_{\text{base}} \quad \text{or} \quad \hat{e} > e_{\text{base}} + \theta_E\, (1 - \beta\, u).$$

5. **Cool-down.** Honour the request only if the cool-down period has elapsed; on acceptance, reset baselines and timestamp.

```
function shouldReplan(metrics, params, state):
    # 1. instantaneous guard
    hitRatio = metrics.hits / (metrics.hits + metrics.misses)
    if hitRatio < 0.20:
        return true

    # 2. eviction delta
    deltaEvict = metrics.evictedKeys - state.prevEvict
    state.prevEvict = metrics.evictedKeys

    # 3. memory pressure
    u = metrics.usedMemory / metrics.maxMemory

    # 4. EWMA updates
    state.ewmaHit = params.alpha * hitRatio + (1 - params.alpha) *
        state.ewmaHit
    state.ewmaEvict = params.alpha * deltaEvict + (1 - params.alpha) *
        state.ewmaEvict

    # predicates
    hitDrop = state.ewmaHit < (1 - params.theta_H) * state.baseHit
    evictSpike = state.ewmaEvict > state.baseEvict +
            params.theta_E * (1 - params.beta * u)

    # 5. cool-down check
    if now() - state.lastReplan < params.cooldown:
        return false

    if hitDrop or evictSpike:
        state.baseHit = state.ewmaHit
        state.baseEvict = state.ewmaEvict
        state.lastReplan = now()
        return true
    return false
```

## D  ADAPTIVE REPLANNING UNDER WORKLOAD SHIFT

### D.1  EXPERIMENTAL SETUP

We concatenate four contiguous blocks to create a single *mixed* evaluation stream of **513 requests**:
MOVIE_DATA[:100] + HQA_DATA[:200] + PQA_DATA[:113] + MOVIE_DATA[100:200].
This exactly mirrors the order used in prior work—100 Movie, 200 HotpotQA, 113 ParallelQA,
100 Movie (phase 2). All runs use a tight 25 % memory budget; the planner is invoked at startup
and whenever the Replan Policy fires (three times in total).

### D.2  INITIAL PLANNER "THOUGHTS" AND PLAN

```
- **Movie_Search**
  - **READ**
  - **Primary Args**: `term`
  - **Output**: **TRANSIENT**
  - **Reason**: The output is based on the content of Wikipedia, which
      can change over time. However, the changes are not immediate and
      the content can be considered relatively stable for a period,
      making it suitable for transient caching.

- **HQA_Search**
```

```
   - **READ**
   - **Primary Args**: `term`
   - **Output**: **TRANSIENT**
   - **Reason**: Similar to `Movie_Search`, the output is based on
       Wikipedia content, which can change over time but is relatively
       stable for a period, making it suitable for transient caching.

 - **PQA_Search**
   - **READ**
   - **Primary Args**: `term`
   - **Output**: **TRANSIENT**
   - **Reason**: Like `Movie_Search` and `HQA_Search`, the output is
       based on Wikipedia content, which can change over time but is
       relatively stable for a period, making it suitable for transient
       caching.

 - **Calculate**
   - **READ**
   - **Primary Args**: `problem`
   - **Output**: **STATIC**
   - **Reason**: The output of a math problem is deterministic and does
       not change over time. Once a problem is solved, the result can be
        cached indefinitely as it will always be the same.
```

The resulting JSON plan assigns a 1h TTL to all three search tools and marks `Calculate` as STATIC.

| Tool | Cacheability | TTL (s) | Notes |
|---|---|---|---|
| Movie_Search | TRANSIENT | 3600 | |
| HQA_Search | TRANSIENT | 3600 | |
| PQA_Search | TRANSIENT | 3600 | |
| Calculate | STATIC | — | deterministic |

Table 4: Initial plan.

### D.3 REPLAN CYCLE

Stars in Fig. 4 of the main paper correspond to three replans. Their LLM "Thoughts" blocks (abridged) and the ensuing deltas are summarised below.

**Replan 1 (after 240 req).** TTL for `HQA_Search` is reduced to 600 s and `Calculate` is removed.

```
 - **Movie_Search**: KEEP  Hit ratio is relatively stable at 74.6%, and
     there is no significant memory pressure.
 - **HQA_Search**: SHORTEN_TTL  Hit ratio is extremely low at 0.7%,
     indicating that the data is not being reused effectively, and there
     is a risk of premature eviction due to high evictions.
 - **PQA_Search**: KEEP  No specific hit ratio data provided, but given
     the similar nature to HQA_Search, it might also have a low hit
     rate. However, without concrete data, it's safer to keep the
     current TTL.
 - **Calculate**: DROP  This tool is marked as STATIC, and it should be
     dropped to free up memory for more frequently accessed data.
 - **Final summary**: shortened TTL for HQA_Search to reduce memory
     footprint and improve cache efficiency."
```

**Replan 2 (after 360 req).** HQA reuse collapses to $< 1\%$, so its TTL is cut to 60 s; other tools are kept as is.

```
- **Movie_Search**: KEEP  Hit ratio is relatively stable at 74.6%, and
    there is no significant memory pressure.
- **HQA_Search**: SHORTEN_TTL  Hit ratio is extremely low at 0.7%,
    indicating that the data is not being reused effectively, and there
    is a risk of premature eviction due to high evictions.
- **PQA_Search**: KEEP  No specific hit ratio data provided, but given
    the similar nature to HQA_Search, it might also have a low hit
    rate. However, without concrete data, it's safer to keep the
    current TTL.
- **Calculate**: DROP  This tool is marked as STATIC, and it should be
    dropped to free up memory for more frequently accessed data.
- **Final summary**: shortened TTL for HQA_Search to reduce memory
    footprint and improve cache efficiency."
```

**Replan 3 (after 480 req).** Evictions spike while HQA remains cold; planner keeps the 60 s TTL, no further changes.

```
- **Movie_Search**: KEEP  Despite a high hit ratio (74.6%), the tool
    is not contributing to memory pressure and can be kept.
- **HQA_Search**: SHORTEN_TTL  The hit ratio is extremely low (0.7%
    and 0% in the latest snapshot), indicating that the data is not
    being reused effectively. Shortening the TTL can help reduce memory
    usage.
- **PQA_Search**: KEEP  The hit ratio is moderate (45.4%), and there
    is no significant memory pressure. Keeping the TTL as is can help
    maintain performance
- **Final summary**: adjusted TTLs for HQA_Search to reduce memory
    footprint and maintain overall cache efficiency.
```

| Iteration | Movie | HQA | PQA | Calculate |
|---|---|---|---|---|
| Initial | 3600 | 3600 | 3600 | Static |
| Replan 1 | 3600 | 600 | 3600 | *Dropped* |
| Replan 2 | 3600 | 60 | 3600 | — |
| Replan 3 | 3600 | 60 | 3600 | — |

Table 5: Plan transitions.

| Tool | Hits | Misses | Hit Rate (%) |
|---|---|---|---|
| **Replan 1** | | | |
| Movie_Search | 356 | 121 | 74.6 |
| HQA_Search | 2 | 278 | 0.7 |
| **Replan 2** | | | |
| Movie_Search | 0 | 0 | 0.0 |
| HQA_Search | 0 | 120 | 0.0 |
| PQA_Search | 83 | 100 | 45.4 |
| **Replan 3** | | | |
| Movie_Search | 212 | 144 | 59.6 |
| HQA_Search | 0 | 0 | 0.0 |
| PQA_Search | 234 | 29 | 89.0 |

Table 6: Per-tool hit rates at each replan point.

| Checkpoint | Evictions | Hits | Misses |
|-----------|-----------|------|--------|
| Replan 1 | 312 | 358 | 399 |
| Replan 2 | 544 | 441 | 619 |
| Replan 3 | 705 | 887 | 792 |

Table 7: Redis stats observed across replan checkpoints.

### D.4 TAKE-AWAYS

1. **Planner reasoning is interpretable.** The free-form "Thoughts" blocks expose why each decision is made, easing manual verification.
2. **Fine-grained TTL tuning matters.** Demoting `HQA_Search` from 3600→600→60 s prevents low-reuse keys from monopolising memory, while hot Movie/PQA results persist.
3. **Cold static tools are expendable.** Dropping `Calculate` sacrificed no reuse but freed 5% of the budget.
4. **Dropped tools are not revisited.** Once a tool is dropped (e.g., for being static and cold), the current planner does not reconsider it—even if memory pressure later eases or the underlying state becomes more dynamic. Supporting tool revival through low-cost probing or recency-based re-injection remains promising future work.

This adaptive loop shows how a lightweight EWMA trigger paired with LLM-driven replanning keeps the cache aligned with shifting locality without manual intervention.

## E DEPENDENCY-AWARE INVALIDATION ON $\tau$-BENCH RETAIL

### E.1 EXPERIMENTAL SETUP

We replay the GPT-4o trajectory for the `retail` subset of $\tau$-bench, which issues **582** tool calls spanning **15** unique tools and 115 high-level tasks. ToolCacheAgent (TCA) generates a single cache plan *a-priori*; All tools run against the benchmark's in-memory mocks with a local Redis backend (10 MB budget).

### E.2 PLANNER INSIGHTS AND FAILURE ANALYSIS

**LLM "Thoughts".** Before emitting the JSON plan, the planner writes a free-form **Thoughts** block that explains every tool decision. An excerpt is shown below (full text in the replication bundle):

```
### Thoughts

- **calculate**
  - **Type**: READ
  - **Primary Args**: `expression`
  - **Output Type**: STATIC
  - **Reasoning**: The result of a mathematical expression is
      deterministic and does not change over time unless the input
      expression changes.

- **cancel_pending_order**
  - **Type**: WRITE
  - **Invalidates**: `get_order_details`
  - **Mapping**: `order_id` maps to `get_order_details`'s `order_id`.

- **exchange_delivered_order_items**
  - **Type**: WRITE
  - **Invalidates**: `get_order_details`, `get_product_details`
```

```
  - **Mapping**: `order_id` maps to `get_order_details`'s `order_id`. `
     new_item_ids` and `item_ids` map to `get_product_details`'s `
     product_id` (assuming item IDs are derived from product IDs).

 - **find_user_id_by_email**
   - **Type**: READ
   - **Primary Args**: `email`
   - **Output Type**: STATIC
   - **Reasoning**: The user ID associated with an email is a static
     piece of information and does not change over time.

 - **find_user_id_by_name_zip**
   - **Type**: READ
   - **Primary Args**: `first_name`, `last_name`, `zip`
   - **Output Type**: STATIC
   - **Reasoning**: The user ID associated with a combination of first
     name, last name, and zip code is a static piece of information
     and does not change over time.

 - **get_order_details**
   - **Type**: READ
   - **Primary Args**: `order_id`
   - **Output Type**: TRANSIENT
   - **Reasoning**: The status and details of an order can change over
     time (e.g., order status, item details, shipping address), but
     the changes are not immediate and can be cached for a short
     period.

 - **get_product_details**
   - **Type**: READ
   - **Primary Args**: `product_id`
   - **Output Type**: TRANSIENT
   - **Reasoning**: The inventory details of a product can change over
     time (e.g., stock levels, prices), but the changes are not
     immediate and can be cached for a short period.

 - **get_user_details**
   - **Type**: READ
   - **Primary Args**: `user_id`
   - **Output Type**: TRANSIENT
   - **Reasoning**: The details of a user, including their orders, can
     change over time (e.g., address, order history), but the changes
     are not immediate and can be cached for a short period.

 - **list_all_product_types**
   - **Type**: READ
   - **Primary Args**: None
   - **Output Type**: STATIC
   - **Reasoning**: The list of product types is a static piece of
     information and does not change over time.

 - **modify_pending_order_address**
   - **Type**: WRITE
   - **Invalidates**: `get_order_details`
   - **Mapping**: `order_id` maps to `get_order_details`'s `order_id`.

 - **modify_pending_order_items**
   - **Type**: WRITE
   - **Invalidates**: `get_order_details`, `get_product_details`
   - **Mapping**: `order_id` maps to `get_order_details`'s `order_id`. `
     new_item_ids` and `item_ids` map to `get_product_details`'s `
     product_id` (assuming item IDs are derived from product IDs).
```

```
  - **modify_pending_order_payment**
    - **Type**: WRITE
    - **Invalidates**: `get_order_details`
    - **Mapping**: `order_id` maps to `get_order_details`'s `order_id`.

  - **modify_user_address**
    - **Type**: WRITE
    - **Invalidates**: `get_user_details`
    - **Mapping**: `user_id` maps to `get_user_details`'s `user_id`.

  - **return_delivered_order_items**
    - **Type**: WRITE
    - **Invalidates**: `get_order_details`, `get_product_details`
    - **Mapping**: `order_id` maps to `get_order_details`'s `order_id`. `
        item_ids` map to `get_product_details`'s `product_id` (assuming
        item IDs are derived from product IDs).

  - **transfer_to_human_agents**
    - **Type**: WRITE
    - **Invalidates**: None
    - **Mapping**: None
    - **Reasoning**: This function transfers the user to a human agent
        and does not affect any state that is read by other tools.
```

**Residual Failures (6 / 582 requests).**   Dependency-aware invalidation left six stale responses:

- **Gift-card balance skew (5×)** – Five get_user_details calls showed an out-of-date gift-card balance after order-state mutations by cancel_pending_order, return_delivered_order_items, or modify_pending_order_items.   The in-validation map for these WRITE tools tracks order_id but does not identifies the dependency with the balance part of the user detail.
- **Stale name–ZIP lookup (1×)** – A single find_user_id_by_name_zip tool call failed after modify_user_address changed the customer's ZIP code.