# OpenReview forum: "Tool Cache Agent: Accelerating LLM Agent Through Intelligent Tool Call Caching"
_ICLR.cc/2026/Conference — ICLR 2026 Conference Withdrawn Submission_

### Official Review · Reviewer_aBdK · 2025-10-31

**Soundness:** 2
**Presentation:** 3
**Contribution:** 2
**Rating:** 2
**Confidence:** 3

**Summary:**

This paper introduces a  framework that reduces redundant computation in LLM agent workflows by caching the results of tool calls. The core is an LLM cache planner that determines the cacheability levels (STATIC/TRANSIENT/NONE) of tools. The system dynamically replans cache strategies based on runtime telemetry such as cache hit rate, eviction events, and memory usage. Experiments across multiple agent architectures (ReAct, LLMCompiler) and datasets (HotpotQA, Movie Recommendation, ParallelQA, τ-bench Retail) demonstrate consistent latency improvements (up to 1.69×) without accuracy loss.

**Strengths:**

1. The paper addresses an important and practical problem, which targets improving the efficiency of agent tool calls. This directly affects both computational cost and real-world latency in LLM agent applications.
2. The proposed approach is conceptually simple yet well-motivated, with a clear and coherent system design.
3. The experimental setup is reasonable, as it evaluates both efficiency gains and the potential side effects of caching, providing a comprehensive analysis.

**Weaknesses:**

1. The evaluation is conducted only on relatively small datasets (HotpotQA, Movie Rec., ParallelQA, and a τ-bench subset), with a small number of tools involved. It would be more convincing to include larger or more diverse tool-calling benchmarks (e.g., BFCL or other complex multi-tool agent suites).
2. Real-world agents often involve multi-step, stateful, or interdependent tool interactions, but the current experiments do not sufficiently stress-test the system in such scenarios.
3. The Cache Planner heavily relies on a large LLM (Qwen 2.5 72B) for reasoning and plan generation, which introduces additional computational overhead and potential variability in performance. Although current results show no accuracy drop, it remains uncertain whether this scales well to more complex or longer-horizon tasks.

**Questions:**

What is the overhead introduced by using Qwen 2.5 72B for cache planning? Has this overhead been included in the overall latency or efficiency measurements reported in the paper?

---

### Official Review · Reviewer_VqNC · 2025-10-31

**Soundness:** 2
**Presentation:** 2
**Contribution:** 1
**Rating:** 2
**Confidence:** 4

**Summary:**

In this paper, the authors introduces ToolCacheAgent, an adaptive framework that reduces redundant tool executions in large language model agents. It automatically learns which tool calls are cacheable, assigns expiration policies, and performs dependency-aware invalidation to maintain correctness when data changes. Experiments across benchmarks show up to 1.69x latency reduction without accuracy loss. ToolCacheAgent establishes caching as a key optimization layer for improving efficiency and scalability in LLM-based agent systems.

**Strengths:**

The proposed method demonstrates up to a 1.69× latency improvement without compromising accuracy, highlighting its effectiveness in optimizing LLM agent performance.

**Weaknesses:**

1. The necessity of a proxy component for caching is unclear. What advantages does it offer over directly storing tool outputs? Moreover, tools with high-frequency variability (e.g., time or date) challenge cache validity—how is this addressed in ToolCacheAgent?
2. Table 1 contains many identical accuracy values. To substantiate the claim that accuracy remains unaffected, a theoretical justification or analysis would strengthen the argument.

**Questions:**

1. How is dataset latency computed—by averaging across all samples?
2. Does the sample order influence latency measurements?

---

### Official Review · Reviewer_FcUJ · 2025-10-31

**Soundness:** 3
**Presentation:** 2
**Contribution:** 3
**Rating:** 4
**Confidence:** 3

**Summary:**

This paper introduces ToolCacheAgent, an adaptive caching framework designed for LLM agents. It reduces redundant computations by intelligently caching tool invocation results, improving response speed without compromising accuracy. Its core purpose is to address the performance bottleneck caused by high tool usage and repeated executions in LLM agents.

**Strengths:**

1. Unlike existing research focused on LLM inference acceleration or parallel tool scheduling, the paper explicitly identifies "redundant tool calls" as a dominant bottleneck in agent workflows. It is the first work to formalize "tool call caching" as a core optimization layer for LLM Agents, filling a key research void. The proposed ToolCacheAgent introduces unique components that go beyond simple caching mechanisms.
2. It integrates LLM reasoning (for cache planning) with Redis-based caching (for efficient storage) and dependency tracking (for correctness), merging strengths of NLP (LLM semantic understanding) and systems (cache management) to solve the problem.
3. Considering the memory pressure and low hit rate present in real-world scenarios, the framework has designed a replan mechanism.
4. For benchmark with a high reuse rate of tool calls, a maximum acceleration of 1.69 times is achieved. This indicates that starting from the issue of high tool call reuse, it can indeed effectively reduce the total response time, and the proposed framework has achieved significant results.

**Weaknesses:**

1. ToolCacheAgent’s adaptive replanning is designed to handle "shifting workloads and memory pressure", but the paper’s experimental setup for workload shifts is overly simplistic and lacks diversity—limiting confidence in the mechanism’s real-world adaptability. Real-world agents often face interleaved queries (e.g., 5 Movie, 2 HotpotQA, 3 ParallelQA, repeated) rather than block-based shifts. The current setup does not test if replanning can handle frequent, small-scale workload changes. Actionable improvement might be applying an interleaved workload block  to test replanning’s responsiveness to frequent, unstructured shifts.
2. The paper does not report cases where the Cache Planner generated invalid plans or how the system recovers.
3. The paper defines four core parameters to govern when adaptive replanning is triggered: \alpha, \theta_H, \theta_E, \beta. These values are used as defaults without further validation. There is no testing of how parameter variations affect performance. Ablation experiments should be conducted to investigate the effects of several key parameters.
4. The article does not provide an analysis of the time consumption for LLMs' planning and replanning, and the experiment only used two LLMs of about 70B in size. Can LLMs of different sizes achieve comparable results? Do smaller LLMs consume less time for planning and replanning? More comprehensive experiments can be conducted on the above aspects.
5. Table 1 only lists two memory budgets (50% and 100%). More memory budgets should be evaluated to better analyze the bottlenecks of memory budgets.

**Questions:**

1. Does the effectiveness of plan and replan depend significantly on the size, architecture of LLMs and the prompt?
2. If this framework were to be deployed in a real environment, how difficult would it be, and what problems might be encountered?

---

### Official Review · Reviewer_ZLwL · 2025-11-03

**Soundness:** 4
**Presentation:** 4
**Contribution:** 2
**Rating:** 2
**Confidence:** 4

**Summary:**

This paper presents ToolCacheAgent, an adaptive framework that caches the results of tool calls. Instead of recomputing identical tool invocations, the system stores outputs based on a dynamic caching plan determined by a Cache Planner—a language model–driven module that classifies tools (READ/WRITE), estimates cacheability (STATIC, TRANSIENT, NONE), and defines invalidation rules. The paper also proposes dependency-aware invalidation to prevent stale reads when related tools modify shared state, and an adaptive replanning mechanism that updates cache policies in response to runtime metrics (hit rates, evictions, and memory pressure).
Empirical results on three benchmarks (HotpotQA, Movie Recommendation, ParallelQA) demonstrate up to 1.69× latency speedups without accuracy loss, and additional experiments show adaptive resilience and correctness benefits from dependency-aware invalidation.

**Strengths:**

+ Dependency-aware invalidation: The parameter-level invalidation mechanism demonstrates a rigorous approach to maintaining cache correctness—a nontrivial issue in stateful tool environments.

+ Empirical validation: Multiple datasets and workloads convincingly show reduced latency and preserved accuracy. The adaptive replanning analysis (e.g., TTL decay, dropping low-utility tools) showcases robustness.

+ Clear contribution separation: The paper cleanly delineates end-to-end caching gains, replanning adaptability, and correctness evaluation, which strengthens the technical narrative.

**Weaknesses:**

Initial Cache Planning Ambiguity – The Cache Planner determines caching strategies during the initial planning phase using only static metadata (tool names, descriptions). However, the mechanism by which it converts such high-level descriptions into concrete cacheability decisions (e.g., STATIC vs. TRANSIENT) is underexplained. Is this accomplished through a single LLM prompt (as hinted in Appendix A) or a rule-based prior?

Guaranteeing Cache Freshness – The dependency-aware invalidation resembles standard memory caching strategies, yet the analogy breaks down because the environment can change independently (e.g., external APIs, search queries). In traditional systems, “dirty pages” have explicit writebacks or coherence mechanisms. In ToolCacheAgent, it remains unclear how the system guarantees correctness when external data sources mutate outside the agent’s observation. The invalidation relies on argument mapping between tools, but cannot handle latent dependencies or background state drift (the paper itself acknowledges this limitation in §5.3). A discussion of staleness guarantees or consistency bounds would be valuable.

Evaluation Limitations – The latency experiments are run on simulated environments (e.g., Redis backend, local datasets), which may underestimate real-world API/network costs.

Table 3 - "ToolCacheAgent successfully identified the role and cacheability class for all 15 tools in the trace". I would like to see how well does this system when you have order of hundreds of tools.

Table 3 - "..Without invalidation, 35 tool calls returned incorrect results; enabling invalidation reduced this to 6—a 5.8×reduction in error rate." Isn't the goal to have 0% error rate for it to be similar to "caching"?

**Questions:**

What are the computational costs (both time and token usage) of running the LLM based Cache Planner and Replanner during deployment?

---

### Note · Authors · 2025-11-12

I have read and agree with the venue's withdrawal policy on behalf of myself and my co-authors.